# Prevalence of Anxiety and Depression Symptoms and Their Relationship with Nutritional Status and Mortality in Patients with Colorectal Cancer

**DOI:** 10.3390/ijerph192013548

**Published:** 2022-10-19

**Authors:** Virginia Soria-Utrilla, Francisco José Sánchez-Torralvo, Iván González-Poveda, Santiago Mera-Velasco, Nuria Porras, José Antonio Toval-Mata, María García-Olivares, Manuel Ruiz-López, Montserrat Gonzalo-Marín, Joaquín Carrasco-Campos, María José Tapia, Julio Santoyo-Santoyo, Gabriel Olveira

**Affiliations:** 1Unidad de Gestión Clínica de Endocrinología y Nutrición, Hospital Regional Universitario de Málaga, 29007 Malaga, Spain; 2Instituto de Investigación Biomédica de Málaga (IBIMA), 29010 Malaga, Spain; 3Departamento de Medicina y Dermatología, Facultad de Medicina, University of Malaga, 29010 Malaga, Spain; 4Unidad de Gestión Clínica de Cirugía General y Digestiva, Hospital Regional Universitario de Málaga, 29010 Malaga, Spain; 5CIBER de Diabetes y Enfermedades Metabólicas Asociadas, Instituto de Salud Carlos III, 28029 Madrid, Spain

**Keywords:** cancer, colorectal, oncology, anxiety, depression, HADS, malnutrition

## Abstract

Background: Anxiety and depression are common in patients with cancer. The aim of this study is to determine the prevalence of anxiety and depression symptoms in colorectal cancer (CRC) patients awaiting elective surgery and whether there is an association with their preoperative nutritional status and postoperative mortality. Methods: A prospective study was conducted on 215 patients with CRC proposed for surgery. Data about nutritional status were collected using the Global Leadership Initiative on Malnutrition (GLIM) criteria, while anxiety and depression symptoms data were collected using Hospital Anxiety and Depression Scale (HADS). Results: HADS detected possible anxiety in 41.9% of patients, probable anxiety in 25.6%, possible depression in 21.9%, and probable depression in 7.9%. GLIM criteria found 116 (53.9%) patients with malnutrition. The HADS score for depression subscale was significantly higher in malnourished patients than in well-nourished (5.61 ± 3.65 vs. 3.95 ± 2.68; *p* = 0.001). After controlling for potential confounders, malnourished patients were 10.19 times more likely to present probable depression (95% CI 1.13–92.24; *p* = 0.039). Mortality was 1.9%, 4,2%, and 5.6% during admission and after 6 and 12 months, respectively. Compared to patients without depressive symptomatology, in patients with probable depression, mortality risk was 14.67 times greater (95% CI 1.54–140.21; *p* = 0.02) during admission and 6.62 times greater (95% CI 1.34–32.61; *p* = 0.02) after 6 months. Conclusions: The presence of anxiety and depression symptoms in CRC patients awaiting elective surgery is high. There is an association between depression symptoms, preoperative nutritional status, and postoperative mortality.

## 1. Introduction

Colorectal cancer (CRC) is the third most common malignancy worldwide and the second leading cause of cancer-related mortality [1,2]. Moreover, it is ranked second to fourth in terms of incidence, showing an increasing trend [3]. Survival of CRC patients has significantly improved in the last few years due to earlier diagnosis and improvements in treatment [4], of which surgery is the primary course in most cases [2]. Nevertheless, as the survival rates of CRC patients rise, new challenges emerge [4]. Many cancer survivors often experience physical and mental health problems that impact their well-being [4,5].

Physical health in patients with cancer depends to a great extent on their nutritional status. Cancer and its surgery involve a systemic inflammatory state that leads to a metabolic stress response with hypercatabolism [1], contributing to malnutrition [6]. In patients undergoing cancer surgery, preoperative malnutrition is, as with any other condition that worsens tolerance to the stress of surgery, a risk factor for poor outcomes [1]. Malnutrition is an independent risk factor for the incidence of complications and increased mortality, hospital length of stay, and costs after surgery. Moreover, malnutrition could lead to a faster progression of the disease and impaired functional status [7]. Unfortunately, malnutrition is common among cancer patients [8], with a prevalence of up to 60% reported in CRC patients [9].

Moreover, depression and anxiety can arise in patients with cancer due to both psychological reactions and direct biological effects of tumors or their treatments [10,11]. Depression and anxiety have a higher prevalence in patients with cancer than in the general population, affecting up to 20% and 10% of cancer patients, respectively [10]. The presence of these mental health problems in cancer patients is associated with reduced functioning, decreased adherence to treatment, lower cancer survival, and higher treatment costs [10,12,13]. However, most of the cancer patients who suffer from psychological problems do not receive any therapy for these conditions [4,10,12].

Both malnutrition and psychological distress are common issues among patients with cancer. However, as far as we know, published data on the prevalence of anxiety and depression symptoms and their relationship with preoperative nutritional status and postoperative mortality in CRC patients awaiting elective surgery are scarce.

Our hypothesis is that the prevalence of anxiety and depression symptoms is high in CRC patients awaiting elective surgery and it is related to a high rate of preoperative malnutrition and postoperative mortality in these patients.

Therefore, the aim of the present study is to determine the prevalence of anxiety and depression symptoms in CRC patients awaiting elective surgery and whether there is an association between anxiety and depression symptoms, preoperative nutritional status, and postoperative mortality in these patients.

## 2. Materials and Methods

This prospective study was performed at Hospital Regional Universitario de Malaga. A total of 215 patients were assessed for eligibility after medical evaluation at the Coloproctology Unit. Inclusion criteria were age ≥ 18 years old, diagnosis of CRC with a proposal for curative surgery between October 2018 and July 2021, and ability to sign the informed consent. The exclusion criterion was a refusal to sign the informed consent. All patients signed the informed consent, so 215 patients were included in the study. Enrollment was carried out before surgery. Data about nutritional status and anxiety and depression symptoms were collected (Figure 1).

### 2.1. Assessment of Nutritional Status

A nutritional assessment according to Global Leadership Initiative on Malnutrition (GLIM) criteria [6,7] was performed one week after the first medical evaluation in the consultation of the Coloproctology Unit. At least one phenotypic criterion and one etiologic criterion must be present for the diagnosis of malnutrition [6].

For the application of phenotypic criteria, we assessed the presence of unintentional weight loss (>5% within past 6 months), low body mass index (BMI) (<20 kg/m^2^ for < 70 years, or <22 kg/m^2^ for >70 years), or reduced muscle mass (fat-free mass index (FFMI) <17 kg/m^2^ for men, or <15 kg/m^2^ for women). To obtain these data, weight was measured with a weighing scale adjusted to 0.1 kg (SECA 665, Hamburg, Germany) and height was measured with a stadiometer (Holtain Limited, Crymych, UK). BMI was calculated from these two variables. Finally, FFMI was obtained using bioelectrical impedance analysis (BIA) (Akern BIA-101/Nutrilab analyzer, Akern SRL, Pontassieve, Florence, Italy).

Regarding etiologic criteria, we considered CRC as a chronic inflammatory condition.

### 2.2. Assessment of Symptoms of Anxiety and Depression

The presence of symptoms of anxiety and depression was measured with the Hospital Anxiety and Depression Scale (HADS), a self-assessment scale valid for screening that reports patients’ symptoms in the past week [14]. This questionnaire is divided into two independent subscales, anxiety and depression, with 7 items for each subscale. Each item must be answered from 0 to 3, so the possible scores range from 0 to 21 for anxiety and from 0 to 21 for depression. A score of 0 to 7 for either subscale is considered as normal, a score of 8 to 10 is suggestive of the presence of the mood disorder (‘doubtful case’), and a score of 11 or higher indicates probable presence (‘caseness’) of the respective state [4,14].

### 2.3. Clinical Outcomes

Data concerning postoperative complications (such as postoperative collection, paralytic ileus, surgical wound infection, suture dehiscence, febrile syndrome, or bleeding) and mortality during admission and after 1, 6, and 12 months were obtained from the review of patients’ medical history. Complications were considered for the study when their degree of severity was 2 or higher in the Clavien–Dindo classification.

### 2.4. Data Analysis

Quantitative variables were expressed as the mean ± standard deviation. Comparison between qualitative variables was conducted via a chi-square test, with Fisher correction if necessary. Quantitative variable distribution was assessed using the Kolmogorov–Smirnov test. Differences between quantitative variables were analyzed using Student’s t test, and for variables not following a normal distribution, using nonparametric tests (Mann–Whitney). Multivariate logistic regression models were used to assess the relationship between the presence of symptoms of anxiety and depression and the diagnosis of malnutrition, controlling also for sex, age, and tumor stage. For calculations, significance was set at *p* < 0.05 for two tails. The data analysis was performed with the SPSS 22.0 program (SPSS Inc., Chicago, IL, USA, 2013).

### 2.5. Ethics

The Provincial Research Ethics Committee of Malaga approved the study (reference number #26072018) and informed consent was obtained from all participants. The ethical principles included in the latest revision of the Declaration of Helsinki and good clinical practice standards were applied.

## 3. Results

A total of 215 patients were evaluated. Their mean age was 68.4 ± 10.3 years. In total, 57.2% were male and 42.8% were female. Their general features are shown in Table 1. Colon cancer was more frequent (57.7%) than rectum cancer (42.3%), and most patients were at stages II and III (75.4%) after postoperative pathological study (Table 1).

HADS presented an average score of 7.38 ± 4.59 points for the anxiety subscale, and an average score of 5 ± 3.5 points for the depression subscale. HADS detected the possible presence of anxiety (HADSA ≥ 8) in 41.9% of patients, and the probable presence of anxiety (HADSA ≥ 11) in 25.6%. Likewise, HADS detected the possible presence of depression (HADSD ≥ 8) in 21.9% of patients, and the probable presence of depression (HADSD ≥ 11) in 7.9% (Table 1).

In the application of GLIM criteria for the diagnosis of malnutrition, we found that 106 patients (49.3%) had lost more than 5% of their weight in the previous six months and 18 patients (8.4%) had a low BMI. Using BIA as a determinant of muscle mass, 21 patients (9.8%) had an FFMI below the cut-off points. With these data, GLIM criteria detected malnutrition in 116 patients (53.9%).

In malnourished patients, the HADS score was significantly higher with respect to depression (5.61 ± 3.65 in malnourished vs. 3.95 ± 2.68 in well-nourished; *p* = 0.001) and was nonsignificantly higher with respect to anxiety (7.57 ± 4.96 in malnourished vs. 6.62 ± 3.94 in well-nourished; *p* = 0.15) (Figure 2).

After adjusting for age, sex, and cancer stage, the risk of probable anxiety was significantly higher in malnourished patients (95% CI 1.04–4.63; *p* = 0.04). Furthermore, malnourished patients were 3.68 times more likely to present the possible presence of depression (HADSD ≥ 8) than well-nourished (95% CI 1.53–8.86; *p* = 0.004), and 10.19 times more likely to present a probable depression (HADSD ≥ 11) (95% CI 1.13–92.24; *p* = 0.039) (Table 2).

Mortality was 1.9%, 4.2%, and 5.6% during admission and after 6 and 12 months, respectively (Table 1). After adjusting for nutritional status, in patients with possible depression (HADSD ≥ 8), mortality risk was 11.38 times greater than in those without depressive symptomatology [95% CI 1.02–126.56; *p* = 0.048] during the admission, 4.25 times greater [95% CI 1.02–17.69; *p* = 0.046] after 6 months, and 3.52 times greater [95% CI 1.02–12.22; *p* = 0.047] after 12 months. Likewise, in patients with probable depression (HADSD ≥ 11), mortality risk was 14.67 times greater than in those without depressive symptomatology [95% CI 1.54–140.21; *p* = 0.02] during the admission, and 6.62 times greater [95% CI 1.34–32.61; *p* = 0.02] after 6 months. No increased risk of mortality was found in patients with symptoms of anxiety. None of the groups showed an increased risk of surgical complications (Table 3).

## 4. Discussion

The presence of anxiety and depression symptoms in CRC patients awaiting elective surgery was high and there was an association between depression symptoms, preoperative nutritional status, and postoperative mortality in these patients in our study. Despite being more prevalent, anxiety did not show a significant association with postoperative mortality.

For anxiety and depression symptoms screening, we used the Hospital Anxiety and Depression Scale (HADS), a valid and reliable self-detection tool [15], which has been widely studied in patients with cancer [16,17], and is considered an appropriate instrument for screening psychological distress in CRC patients [18]. In fact, a previous study that performed the diagnosis of depression in two steps (first, HADS; then, the major depression section of the Structured Clinical Interview for the DSM-IV) showed a prevalence of major depression of 7% in CRC patients [10,12], very similar to our results using only HADS screening.

We found a greater impact of depression compared to anxiety in our patients, despite being less prevalent. A possible explanation is that anxiety is an initial reaction to uncertainty at the time of cancer diagnosis, but tends to decrease once patients become familiar with their oncological process, treatments, and prognosis, whereas depressive symptoms are more stable over time [13]. In fact, the 2017 WHO World Health Survey showed that, for people with one or more chronic physical diseases and comorbid depression, it was depression that had the greatest effect on worsening health scores and increasing disability, compared to the other chronic diseases [11].

For nutritional assessment, we used GLIM criteria, which were born as a result of a consensus of several international organizations for a definition of malnutrition [6]. Several studies reported malnutrition rates in CRC patients from 20% to 37%, depending on the tool used to assess nutritional status [1,19,20]. Other studies showed that malnutrition can affect up to 60% of these patients [21,22]. In fact, a very recent study informed that 60.68% of CRC patients were malnourished according to GLIM criteria [9], more similar to our results using the same criteria, although slightly higher, maybe due to its higher percentage of patients with advanced tumor stages. Moreover, the prevalence found of malnutrition in patients with cancer is higher when using GLIM criteria rather than other criteria such as the previous ESPEN diagnostic criteria [23].

We found an association between depressive symptomatology and malnutrition, stronger than that found between anxiety and malnutrition. Previous studies also point towards this association. Tiblom et al. used both GLIM criteria and the HADS scale to study malnutrition in relation to anxiety and depression in 273 patients with head and neck cancer, finding that malnourished patients scored significantly worse in depression compared to well-nourished, with no significant differences for anxiety [15]. Nevertheless, Chabowski and Chenjing Zhu reported that a better nutritional status was significantly associated with lower levels of both anxiety and depression assessed by the HADS scale in patients with lung cancer [16,24]. These different results are probably due to the different scales used for the diagnosis of malnutrition and the inclusion of different types of cancer. For CRC patients in particular, Gillis et al. collected 266 patients from five prehabilitation trials before CRC surgery and observed that HADSD was significantly worse as nutritional status worsened, with no significant differences for HADSA [1]. In any case, many studies highlight the importance of depression in CRC patients even though anxiety is more frequent. For instance, Aminisani et al. found a higher frequency of probable and possible anxiety (31.8% and 14%, respectively) than probable and possible depression (20.4% and 17.2%, respectively), although depression had a stronger impact on the overall quality of life [18].

We also studied whether the presence of depression symptoms before surgery is associated with postoperative complications. In our study, we did not perform psychological intervention when we detected depressive symptomatology (HADSD ≥ 8), although we believe it would be convenient. In fact, several studies use the concept of trimodal prehabilitation, which adds psychological support to nutritional therapy and physical exercise training before surgery, to improve surgical outcomes [7,20]. However, a meta-analysis and a clinical trial that tried to determine the impact of trimodal prehabilitation showed no effects on postoperative complications after CRC surgery [25,26]. Consistent with these results, we also found no association between depression symptoms before surgery and postoperative complications, perhaps because they are more dependent on the surgical technique and functional status [22]. A more recent systematic review and meta-analysis reported that trimodal prehabilitation not only did not reduce postoperative complications but also did not improve postoperative mortality or readmission rates [27]. However, psychological interventions could be positive in other aspects, such as reducing length of hospital stay [25,28] or improving postoperative recovery [20].

We found an association between depressive symptomatology and mortality, even after adjusting for nutritional status. A systematic review and meta-analysis published in 2019 reported that depression predicted poorer survival in patients with cancer [29]. A cohort study published in 2021 analyzed data from patients with different types of cancers, including 2807 CRC patients, who had been screened for major depression as part of their cancer care, and also found that major depression was associated with worse survival in all the cancer types they studied, including CRC (HR 1.47, 95% CI 1.11, 1.94) [30]. However, it remains unclear how depression and anxiety impact cancer outcomes [29]. Endocrinological and immunological pathways have been proposed to explain the relationship between psychological distress and survival in cancer patients [31,32]. Moreover, other elements may be of importance in the long term. Saini et al. also concluded that anxiety was twice as frequent as depressive symptoms, but they did not find a significant increase in mortality in cancer patients with anxiety or depressive symptoms at a 13-year follow-up [32].

However, depression in patients with cancer usually remains undetected [11]. A Scottish study that performed routine screening for depression in cancer patients showed that 73% of depressed cancer patients were not receiving any treatment for depression, 24% were receiving an antidepressant drug, and only 5% were visiting a mental health professional [12]. In CRC patients in particular, Matsushita et al. found that only 8% of CRC patients with depression symptoms were seeking treatment [33]. For this reason, routine screening for depression in CRC patients should be performed for adequate management of these patients [11].

Our study has several strengths. It was a prospective study with a sufficient sample size to ensure adequate statistical power. Moreover, most of the tools we used in our study can be easily employed in both outpatient and hospitalized patients. Moreover, our results are based on long-term monitoring. Finally, consecutive recruitment eliminates selection bias.

Yet, this study has some limitations. It was a single-center, observational study, so the results should be interpreted with caution. Socioeconomic factors that could interfere (e.g., educational level, employment status, or marital status) were not taken into account. Furthermore, the data collected about emotional distress do not provide information on changes over time, nor do they establish a diagnosis of anxiety or depression. Moreover, the associations found in our study do not necessarily imply causality.

The high prevalence of anxiety and depression symptoms found in our study and the relationship found between depression symptoms, preoperative nutritional status, and postoperative mortality in CRC patients awaiting elective surgery are robust arguments for the implementation of systematic screening for psychological distress and, if necessary, providing comprehensive psychosocial support to these patients, not forgetting the need for a correct malnutrition screening for adequate nutritional counseling and support.

## 5. Conclusions

The presence of anxiety and depression symptoms in CRC patients awaiting elective surgery is high. There is an association between depression symptoms, preoperative nutritional status, and postoperative mortality. The significant levels of depression and anxiety in these patients indicate the need for a multidisciplinary approach before surgery, as complete and early as possible. Further studies are needed to better understand the causality of this association.

## Figures and Tables

**Figure 1 ijerph-19-13548-f001:**
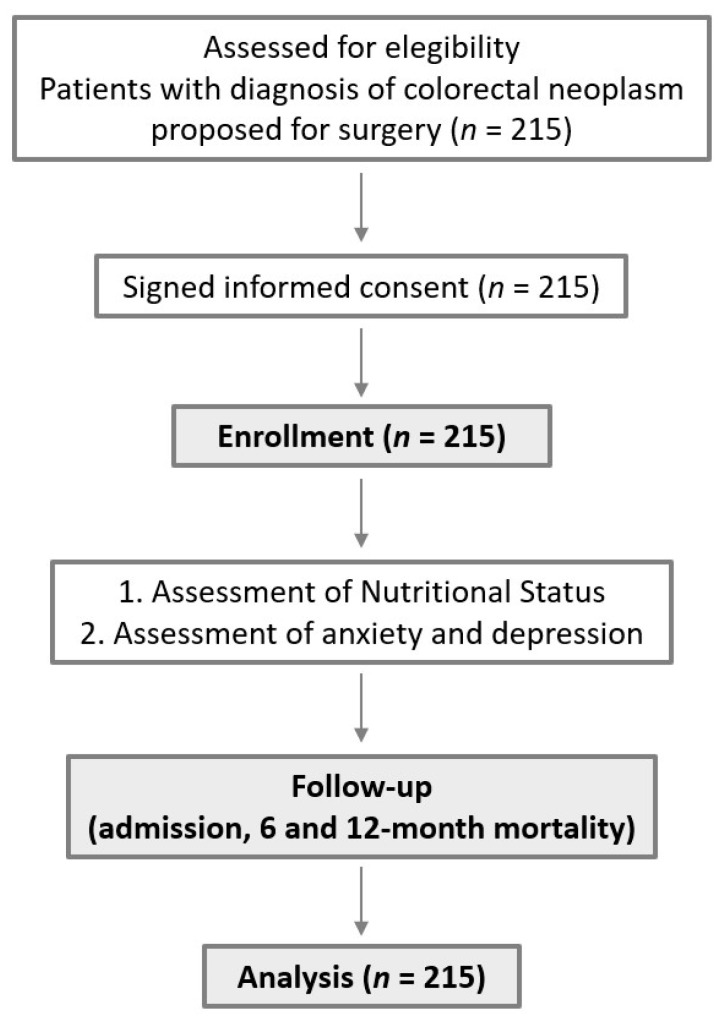
Study flow diagram.

**Figure 2 ijerph-19-13548-f002:**
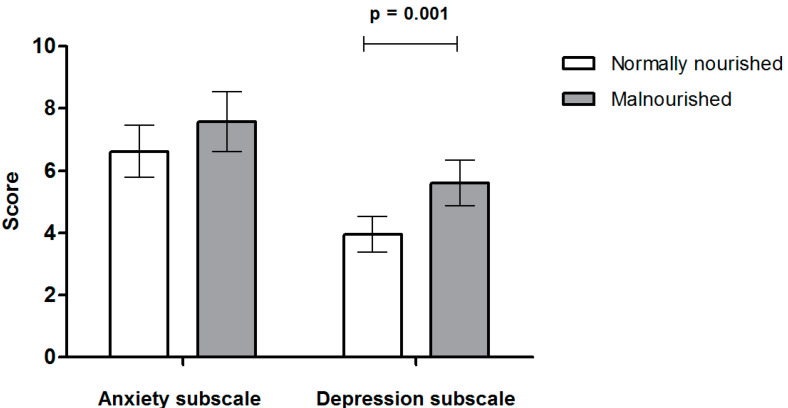
Association between malnutrition and HADS anxiety and depression subscales.

**Table 1 ijerph-19-13548-t001:** General features.

		n = 215
Age (years)	mean ± SD (min–max)	68.4 ± 10.3 (40–89)
Sex	n (%)	
Men		123 (57.2)
Women		92 (42.8)
Type of cancer	n (%)	
Colon		124 (57.7)
Rectum		91 (42.3)
Stage	n (%)	
I		28 (13)
II		67 (31.2)
III		95 (44.2)
IV		25 (11.6)
BMI (kg/m^2^)	mean ± SD (min–max)	26.9 ± 5.2 (15.8–47.6)
Surgical complications	n (%)	90 (41.9)
Malnutrition according to GLIM criteria	n (%)	116 (53.9)
HADSA score	mean ± SD	7.38 ± 4.59
HADSD score	mean ± SD	5 ± 3.59
Possible presence of anxiety (HADSA ≥ 8)	n (%)	90 (41.9)
Possible presence of depression (HADSD ≥ 8)	n (%)	47 (21.9)
Probable presence of anxiety (HADSA ≥ 11)	n (%)	55 (25.6)
Probable presence of depression (HADSD ≥ 11)	n (%)	17 (7.9)
In-hospital mortality	n (%)	4 (1.9)
1-month mortality	n (%)	5 (2.3)
6-month mortality	n (%)	9 (4.2)
12-month mortality	n (%)	12 (5.6)

Abbreviations: BMI = body mass index; GLIM = Global Leadership Initiative on Malnutrition; HADSA = Hospital Anxiety and Depression Scale Anxiety subscale; HADSD = Hospital Anxiety and Depression Scale Depression subscale; SD = standard deviation.

**Table 2 ijerph-19-13548-t002:** Risk of anxious or depression symptomatology according to malnutrition. Adjusted for age, sex, and cancer stage.

	Crude				Adjusted			
	Odds Ratio	95% CI		*p* Value	Odds Ratio	95% CI		*p* Value
		Lower	Upper			Lower	Upper	
Possible presence of anxiety (HADSA ≥ 8)	1.15	0.64	2.06	0.644	1.25	0.66	2.36	0.495
Possible presence of depression (HADSD ≥ 8)	4.45	1.92	10.29	<0.001	3.68	1.53	8.86	0.004
Probable presence of anxiety (HADSA ≥ 11)	1.85	0.93	3.68	0.078	2.19	1.04	4.63	0.04
Probable presence of depression (HADSD ≥ 11)	11.34	1.44	89.08	0.004	10.19	1.13	92.24	0.039

HADSA = Hospital Anxiety and Depression Scale Anxiety subscale; HADSD = Hospital Anxiety and Depression Scale Depression subscale.

**Table 3 ijerph-19-13548-t003:** Risk of surgical complications and mortality according to anxious or depression symptomatology. Adjusted for nutritional status.

		Crude				Adjusted			
		Odds Ratio	95% CI		*p* Value	Odds Ratio	95% CI		*p* Value
			Lower	Upper			Lower	Upper	
Possible presence of anxiety (HADSA ≥ 8)	Surgical complications	1.05	0.61	1.82	0.86	0.94	0.52	1.69	0.83
	Mortality during admission	1.40	0.19	10.11	1	1.52	0.21	11.07	0.68
	Mortality within 6 m	1.12	0.29	4.28	1	1.21	0.31	4.70	0.78
	Mortality within 12 m	0.68	0.20	2.33	0.77	0.74	0.21	2.55	0.63
Possible presence of depression (HADSD ≥ 8)	Surgical complications	1.50	0.78	2.90	0.23	1.43	0.69	2.99	0.34
	Mortality during admission	11.32	1.15	111.48	0.01	11.38	1.02	126.56	0.048
	Mortality within 6 m	4.85	1.25	18.86	0.013	4.25	1.02	17.69	0.046
	Mortality within 12 m	3.93	1.20	12.81	0.016	3.52	1.02	12.22	0.047
Probable presence of anxiety (HADSA ≥ 11)	Surgical complications	0.93	0.50	1.75	0.83	0.79	0.39	1.59	0.52
	Mortality during admission	2.98	0.41	21.69	0.27	2.91	0.39	21.69	0.29
	Mortality within 6 m	1.48	0.37	6.36	0.69	1.41	0.33	5.95	0.64
	Mortality within 12 m	0.97	0.25	3.71	1	0.92	0.24	3.61	0.91
Probable presence of depression (HADSD ≥ 11)	Surgical complications	1.60	0.59	4.33	0.35	1.68	0.53	5.34	0.38
	Mortality during admission	13	1.71	98.89	0.007	14.67	1.54	140.21	0.02
	Mortality within 6 m	6.82	1.54	30.22	0.004	6.62	1.34	32.61	0.02
	Mortality within 12 m	4.48	1.08	18.43	0.025	4.35	0.96	19.66	0.056

HADSA = Hospital Anxiety and Depression Scale Anxiety subscale; HADSD = Hospital Anxiety and Depression Scale Depression subscale.

## Data Availability

Not applicable.

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
