# Peer review of "Prevalence of Anxiety and Depression Symptoms and Their Relationship with Nutritional Status and Mortality in Patients with Colorectal Cancer"

_ijerph, 2022, doi:10.3390/ijerph192013548_

Round 1
Reviewer 1 Report
This is a single-centre observational prospective study assessing the prevalence of anxiety, depression and malnutrition in CRC patients undergoing a curative (?) operation and evaluating any correlation between the three characteristics and postoperative mortality and morbidity.
The article is well-written and easy to read, the study's design is appropriate, the results are well-explained, and the conclusions are consistent with the findings.
But in my opinion, it requires some amendments to make it clearer and more consistent:
1) Materials and methods (M&M), lines 85-86: Please properly state inclusion and exclusion criteria. This point is critical. What do you mean by "patients with a diagnosis of CRC proposed for intervention"? What do you mean by "intervention"? Surgery with an intention to treat? Were palliative surgeries considered (e.g. stoma formation)? Do you mean any other kind of intervention, such as chemotherapy (for stage IV) or (chemo)radiotherapy (for rectal cancer)? Please clearly specify the inclusion and exclusion criteria and the precise time for patient enrollment (e.g. at the diagnosis, before the operation,...).
2) Line 92 in M&M: GLIM criteria, please extend the acronymous here because it was only in the abstract, not in the main text.
3) Line 117-120, Clinical outcomes in M&M: please explain how you assessed the complications. This point is not very clear. Did you consider complication a Clevien-Dindo grade higher than a particular grade? Was every event (even with a CD 1) considered a complication? Please explain better what you considered for complication.
5) Results: You should state in the results how many consecutive patients with the inclusion criteria characteristics did not sign the consent form and missed to the study in the same study period. This is another important point because patients with depression and anxiety could not sign the consent form and be missed. Please discuss this also in the limitations of the study in the discussion paragraph.
4) Line 145, results: please specify if the stage is referred to as the pathological one (postoperative).
5) Table 1 and results: when you refer to the possible presence of anxiety or depression, you state "(HADSA/D ≥8)". This can bring confusion. Are you also including the patients with probable anxiety and depression (≥11)? If not, you should write "(HADSA/D ≥8 and <11)" or (HADSA/D 8-10).
6) Table 3: p-values are not consistent with the text. Especially those of the HADSD≥11. Please amend.
I will be happy to revise this article again when all the points are addressed.
Thank you and best wishes.
Reviewer 2 Report
The proposed work aimed to determine the prevalence of anxiety and depression symptoms in CRC patients awaiting elective surgery. The subject is interesting and a valuable contribution to the literature. Despite the notion that the work is well-described and provides valuable findings, there are a few aspects that should be taken into consideration before publishing in IJERPH. My specific comments are given below:
· There are plenty of grammatical errors along with sentence structuring throughout the manuscript which requires a professional to fix it.
· Although the introduction is well-written however the only lacking point is that the authors have not reported previous studies to build their arguments. Please strengthen the literature section within the introduction.
· The next weak point is the contribution is not well described. Please rewrite to make it impactful.
· How the selected sample size is supposed to be reliable? Please elaborate.
· The discussion is well-described. The authors should add more recent references to make the discussion stronger and more impactful.
· There are a few in-text citations that are not present in the references. Please revise them.
GOOD LUCK
Round 2
Reviewer 1 Report
Dear authors
Thank you very much for having addressed the questions.
After your reply that patients undergoing both curative and palliative surgeries were included, I have some concerns that the relation of causality between depression and mortality is true. The higher mortality rate in depressed and malnourished people may be because more people with advanced disease are depressed and malnourished. So in my opinion you should: 1) state how many patients went for an intention-to-treat operation and how many went for a palliative surgery (stoma formation? resection?); 2) adjust your analysis for the preoperative disease stage.
If you do not do these two amendments, this is a strong limitation to your study, and you should properly state it in the discussion and change the conclusions accordingly (both in the abstract and in the text.
Please check for minor spelling errors.
I will be available to revise the revised version of this manuscript.
Thank you
Best wishes
Author Response
Dear Reviewer,
Thank you for allowing us to improve our article “Prevalence of anxiety and depression symptoms and their relationship with nutritional status and mortality in patients with colorectal cancer.”
In the first answer of the previous report, we made a mistake due to a lack of understanding between us. I personally carried out the study design and ONLY patients with CURATIVE intent were included. As can be seen in Table 1, most of them were in stages II-III. Stage IV patients had single metastases, also operable.
We have changed the error in the manuscript.
Mortality, although it did not present high figures in the sample, precisely because it is an early stage of the disease, was not significantly associated with the stage.
We hope that this clarification resolves the doubts and we apologize for the mistake
We have once again carried out an in-depth check to correct spelling and grammar errors.
Once again, we very much appreciate all the work with the review.
Yours sincerely,
Dr. Francisco José Sánchez Torralvo
Dr. Gabriel Olveira